# FASDQ: Fault-Tolerant Adaptive Scheduling with Dynamic QoS-Awareness in Edge Containers for Delay-Sensitive Tasks

**DOI:** 10.3390/s21092973

**Published:** 2021-04-23

**Authors:** Ruifeng Wang, Ningjiang Chen, Xuyi Yao, Liangqing Hu

**Affiliations:** School of Computer and Electronic Information, Guangxi University, Nanning 530004, China; 1813302007@st.gxu.edu.cn (R.W.); 1913301047@st.gxu.edu.cn (X.Y.); hulq@gxu.edu.cn (L.H.)

**Keywords:** edge container, fault-tolerant scheduling, quality of service, resource management

## Abstract

As the requirement for real-time data analysis increases, edge computing is being implemented to leverage the resources of edge devices to reduce system response times and decrease the latency. However, due to the resource constraints of edge clouds, edge servers are more prone to failures than other systems. Therefore, guaranteeing the reliability of services in edge clouds is critical. In this paper, we propose a fault-tolerant adaptive scheduling mechanism with dynamic quality of service (QoS) awareness (FASDQ), which extends the primary/backup (PB) model by applying QoS on demand to task copies. The aim of the method is to reduce the latency and achieve reliable service for tasks by changing the execution time of task copies. This paper also proposes a container resource-adaptive adjustment mechanism, which adjusts the timing of resources when the available resources cannot meet the task copy requirements. Finally, this paper reports the results of simulation experiments on the EdgeCloudSim platform to evaluate the difference in performance between FASDQ and other methods. The results show that the mechanism effectively reduces the execution time of task copies and outperforms other methods in terms of reliability and general resource utilization.

## 1. Introduction

The goal of edge computing [1] is to obtain real-time responses and provide storage and computation at the edge of the network. This process extends cloud resources to the edge of the network and reduces the network access latency. Although edge computing has many advantages, due to the increased functionality and real-time requirements of smart devices, the chances of resource failures in edge computing are not low, which can reduce the availability of resources and affect the response latency. In smart car networking, smart manufacturing, driverless and other smart devices, and latency-sensitive applications, delays due to reliability issues must be taken seriously. Few previous works have deeply studied the fault tolerance of edge clouds, and tasks are often resubmitted to available resources after they fail, which may lead to unfulfilled service level agreements (SLA) due to the delays encountered by requests [2]. Fault-tolerant scheduling is also widely used as an effective way to assure reliability. However, the current literature on fault-tolerant task scheduling does not sufficiently consider the varying requirements of different tasks for quality of service (QoS) so that edge resources can be allocated as needed. Therefore, the ability to guarantee the reliable completion of latency-sensitive tasks within time constraints and the effective utilization of edge resources is a key issue.

Due to the long startup time of traditional virtual machines and their tendency to waste resources, many systems and studies have used container technology to build the underlying technology stack of edge computing platforms based on the reliability and bandwidth limitations of edge networks [3]. For example, Google launched “the Anthos for Telecom” [4], which will bring Anthos to the edge network by packaging its applications into containers that can run at the edge, providing an immersive experience to customers. Flynn [5] and Deis [6] are also container-based PaaS (platform as a service) projects that can automatically build and deploy any application to a Docker container cluster. Edge Cloud Kubernetes Engine (ECK) [7], launched by Tencent Cloud, and “ACK Edge Kubernetes” [8], launched by AliCloud, ensure that the node’s services are still available in the case of disconnection. Compared with cloud containers [9], edge containers have the advantages of low latency and scalability and are suitable for real-time or latency-sensitive tasks. Limited resources and low-latency services require more in-depth optimization in the application of task scheduling and resource management in edge containers.

It is critical to guarantee the correctness and real-time performance of results in the required timeframe for latency-sensitive tasks. Fault tolerance can be achieved by scheduling multiple copies of a task on different processors. The primary/backup (PB) model [10] and the triple-mode redundancy (TMR) approach [11] are two basic methods in which all copies of a task must be scheduled and generate correct results before its deadline, even if there are errors. In TMR, multiple copies are usually run, and error checking is implemented by comparing the results after completion. With the PB approach, a backup copy is activated if the result of the primary copy is incorrect. The PB model has the advantage of low hardware resource requirements, but it cannot protect against faults [10].

We use the traditional PB model for edge failures and build an edge computing system based on the characteristics of containers. Unlike the traditional cloud environment, in the edge container context, the effective coordination of efficiency and fault tolerance is a challenge. The main contributions of this paper are the following:For latency-sensitive tasks, we propose a dynamic QoS-aware fault-tolerant task scheduling mechanism, which extends the traditional PB model by introducing different reliability requirements for tasks; i.e., when scheduling task copies, their QoS levels are dynamically adjusted according to their reliability requirements and latency constraints.After the primary succeeds, the resources and time interval occupied by the backup are released. In order to make full use of the released resources, the start time of task copies that are waiting in the queue will be reselected to improve resource utilization at the container level.To reduce the resource pressure exerted by the PB model at the edge, we propose a container resource-adaptive adjustment mechanism for the resource quota of each edge container, which replenishes resources according to their usage by the assigned task copies, as well as the latency constraints and reliable service requirements of unexecuted task copies.

The rest of this paper is organized as follows: Section 2 briefly analyzes the problem and related work, and Section 3 provides a systematic analysis. Section 4 describes the FASDQ solution, the primary/backup scheduling algorithm, and the container resource-adaptive adjustment algorithm. Section 5 reports and analyzes the experimental results, and conclusions are drawn at the end of the paper.

## 2. Related Work

Among fault-tolerant approaches, the PB model has been extended to remote clouds in several studies, in which each task has a primary copy and backup copy, and fault-tolerant scheduling is performed in two different computing instances [12]. Primary copies are used when tasks are executed normally, and backup copies are executed when the primary copies fail in order to ensure that the task requests are completed within the specified deadline [13]: this is known as a passive backup scheme [14]. However, the passive backup scheme requires the task to have a sufficient time delay to start the backup copy, which is unrealistic for delay-sensitive tasks. In contrast, an active backup scheme [15] can complete tasks with less slack. The authors of [16] proposed a backup overlay in which task copies are overlapped in the same time slot of the same computation instance, and a release technique, which refers to releasing the resources of the corresponding backup copy after the primary copy succeeds. In [17], two backup schemes were analyzed, and the fault-tolerant elastic scheduling algorithm (FESTAL) was designed for real-time tasks in a virtual cloud environment. This method balances fault tolerance and high resource utilization in the cloud.

Various approaches have been proposed and studied to deal with fault tolerance in cloud computing systems. In edge networks, unlike cloud computing environments, the occurrence of failures may cause a large portion of edge-side resources to go offline, resulting in delays and affecting the overall QoS level. To address the failure of edge servers, redundancy-based replication techniques have also been studied for fault tolerance in edge computing. The research work in [18] provided fault tolerance for the Internet of Things (IoT) by modeling the failure behavior of the Internet grid storage area network (SAN) to provide fault tolerance through redundancy to reduce system downtime. In [19], the PB model was extended to the edge cloud, and a fault-tolerant based QoS-aware scheduling algorithm (FTBQA) was proposed to improve the reliability of edge cloud application services. DeCoRAM middleware [20] tolerates processor failures by considering the execution time and failover order of primary and backup copies in order to meet the soft real-time requirements of each task. To achieve fault tolerance, the authors of [21] proposed a reliable and fault-tolerant agent-based hierarchical IoT–cloud architecture by replicating data at the edge of the network, which redirects the backup server when the service fails to apply. The system in [22] adaptively replicates data according to application requirements for data loss and latency. The replication operation is deferred to some extent, and the data loss and latency requirements are satisfied. In [23], the same input sequence was enforced for virtual machines (VM) replicated across hosts, enabling fast multicore scalable VM fault tolerance. NiLiCon [24] provides seamless failover from a failed container to a backup container of another host, providing fault tolerance to protected applications and external clients. Replication-based fault-tolerant scheduling techniques have been extensively studied in edge cloud environments, but they can also place a burden on resource-limited edge servers to provide timely service to task copies.

Optimizing resource utilization on a fault-tolerant basis is a challenge due to the cost of coordinating reliability safeguards, especially since replication-based fault-tolerance techniques suffer from relatively high resource consumption. Currently, edge computing is still in its infancy in terms of resource management. The authors of [25] developed a computing resource management algorithm for multi-user mobile edge computing (MEC) systems. This algorithm minimizes the average power consumption of mobile devices and MEC servers. Another study [26] proposed a node resource reallocation mechanism based on the characteristics of containers; this method calculates the resource quota for each task in the subsequent phases of task execution and reallocates CPU (central processing unit) resources in the node, making it possible to increase the number of tasks processed in a certain time cycle and effectively reducing the task latency. To some extent, these resource management techniques allocate resources to tasks and effectively improve system resource utilization, but none of them involve fault tolerance. Thus, they cannot support fault tolerance when optimizing resource management.

In summary, for resource-limited edge computing, it is crucial to realize the fault-tolerance capability of the system while effectively utilizing the limited resources at the edge. We find that existing research lacks studies in this area. In this paper, we propose a fault-tolerant adaptive scheduling mechanism with dynamic QoS awareness, which extends the PB model by determining the QoS level for task copies according to reliable service and latency requirements. In addition, during scheduling, FASDQ adaptively adjusts resources according to the resource usage to meet the demands of task copies. This mechanism achieves fault tolerance, accelerates the execution time of tasks, and improves the efficiency of resource use.

## 3. Problem Analysis and System Model

### 3.1. Problem Analysis

Traditional cloud-based industrial Internet of Things (IIoT) architectures struggle to meet requirements such as low latency and high reliability. For example, traditional telecom networks are unable to meet industry-specific requirements for reliable, predictable, and efficient communications. With low latency, scalability, and reduced bandwidth, edge containers are suited for serving real-time or latency-sensitive applications. In 2019, Edgegap, a disruptive gaming startup, released a gaming platform that reduced latency by 58% using real-time telemetry and edge containers [27]. However, edge containers are deployed close to users, their resources are constrained by the environment, and they have higher uncertainty than traditional cloud containers, so reliability assurance for edge containers is critical.

Fault-tolerant scheduling is an effective method to guarantee reliability and is based on replication techniques. The PB model is widely used for fault-tolerant scheduling. In order to reduce the extra resource consumption of backups, the authors of [16] studied a backup overlapping technique in which backups are able to overlap in the same time slots of the same compute instances. However, since the actual compute instances of the edge containers are containers, it is only the primary copies of tasks that cannot be scheduled in the same edge node, whereas their backup copies can overlap in the same container. Another typical technique of the PB model is the reclaiming of resources for backup copies when corresponding primary copies succeed [10]. If the time slot can be utilized after the backup is released, it greatly reduces the latency.

In addition, to meet the strict QoS requirements imposed by latency-sensitive tasks, the method in this paper improves the reliability of task copies by dynamically assigning them QoS levels when task scheduling is performed. Regardless of whether the fault-tolerant technique is based on replication or the assignment of QoS levels to tasks to guarantee reliability, we find that: (1) while processing task requests, task copies are always gathered in containers with higher processing capability, resulting in an uneven distribution of idle and busy containers (in Figure 1, containers C_12_ and C_21_ have more task copies, while container C_31_ is idle); and (2) the existing resources cannot meet the demand of some task copies. Most previous studies have focused on migrating containers as a whole, which has some effect on resource tuning in edge nodes, but they have ignored the effective utilization of resources within individual containers.

Therefore, our method makes more adaptive adjustments to resources at the container and node levels separately. When the resources of a task’s backup are released, the copies waiting in that container are rescheduled so that the container can handle more task copies; if the current resources cannot meet the task demand, nearby edge nodes that are in the shutdown state will be assessed to determine whether they can be turned on. Furthermore, if some resources are idle for a long time during system operation, the resource adjustment mechanism will eliminate the idle resources and thus improve resource utilization.

In the method described in this paper, fault-tolerant adaptive scheduling is performed for latency-sensitive tasks in the edge container, extending the PB model by determining QoS levels for the reliability assurance of task copies. Then, the on-demand adjustment of edge container resources occurs according to resource usage during task scheduling. The approach proposed in this paper not only achieves a tradeoff between the real-time requirement and reliability of tasks, but also avoids reducing resource utilization due to an unbalanced load on edge container resources.

### 3.2. System Model

In Figure 2, edge users can access the edge cloud through wireless networks and submit requests to it. Each edge server includes two modules: the Task Scheduler and the Resource Manager. The Task Scheduler determines whether the edge container can satisfy the requested latency constraint based on the arrival order of requests, and if the request is accepted, the Task Scheduler transmits it to the Resource Manager, which receives tasks and allocates resources for task requests.

Before introducing relevant concepts, we first define the parameters and symbols that are used in this paper, as shown in Table 1.

Relevant concepts and definitions are given below.

There is a set $EN=\left\{EN_{1},EN_{2},\cdots ,EN_{n}\right\}$ of edge nodes around devices, where each edge node has fixed computing resources, including CPU, RAM (random access memory), bandwidth, etc. The processing capability $P_{i}$ (MIPS, million instructions per second) is used to determine the type of computing resources of the edge nodes. Each node $EN_{i}$ includes multiple containers denoted by a set of containers $c_{i}=\left\{c_{i1},c_{i2},\cdots ,c_{im}\right\}$ with different processing capability $P_{ij}$.

We use the set $T=\left\{t_{1},t_{2},\cdots ,t_{n}\right\}$ of latency-sensitive tasks in end devices. Each task has three attributes, $t_{i}=\left\{a_{i},d_{i},l_{i}\right\}$: arrival time $a_{i}$, deadline $d_{i}$, and task size $l_{i}$, which is measured in millions of instructions (MI).

Each task is replicated into two copies: $t_{i}^{P}$ and $t_{i}^{B}$, which are available in different nodes so that the backup can take over execution in another node if the primary fails.

**Definition** **1.**
*Backup states*
$state\left(t_{i}^{B}\right)$
*: the backup is divided into two standby states according to its start time and the completion time of its primary (active (represented by 1) and passive (represented by 0)), as shown in Equation (1).*



(1)$$state\left(t_{i}^{B}\right)=\left\{\begin{array}{ll} 1 & f_{i}^{P}>s_{i}^{B}\\ 0 & otherwise \end{array}\right.$$


If the completion time of the primary copy is later than the start time of its backup copy, the backup state is set to 1; otherwise, it is set to 0. Active backups do not stop until the primary is completed, while passive backups start only after the primary fails. As shown in Figure 3, $t_{1}^{B}$ is passive, but $t_{2}^{B}$ is active, and the part that is redundant with the primary is denoted by $t_{2}^{BR}$.

In container $c_{kl}$, the execution time $e_{kl}\left(t_{i}^{*}\right)$ of $t_{i}^{*}$ (* denotes two copies of tasks: e.g., $t_{i}^{*}$ represents $t_{i}^{P}$ and $t_{i}^{B}$) is the ratio of its task size over the processing capability of this container [28,29]:(2)$$e_{kl}\left(t_{i}^{*}\right)=\frac{l_{i}}{P_{kl}}$$

In this paper, Equation (2) is improved to reduce task latency (see Equations (3) and (5)). The execution time of task copies is changed by assigning different QoS levels that provide different processing capability to different task copies on demand, so the primary copy can be completed as early as possible within the time constraint. In the container $c_{kl}$, the QoS level of $t_{i}^{*}$ is denoted by $q\left(t_{i}^{*}\right)$; $P_{kl}\left(q\left(t_{i}^{*}\right)\right)$ is the processing capability of $t_{i}^{*}$ within $q\left(t_{i}^{*}\right)$. The two copies of a task can be assigned different QoS levels to improve task schedulability. Their execution times and constraint conditions are described below.

**Definition** **2.***The execution time of*$t_{i}^{P}$. *This is defined as the ratio of the task copy size over the processing capability of a container with the QoS level*$q\left(t_{i}^{P}\right)$*, as shown in Equation (3).*


(3)$$eq_{kl}\left(t_{i}^{P}\right)=\frac{l_{i}}{P_{kl}\left(q\left(t_{i}^{P}\right)\right)}$$


In order to meet the delay requirements of primary copies, the following constraints should also be met:(4)$$\forall t_{i}\in T,eq_{kl}\left(t_{i}^{P}\right)\leq d_{i}-est_{i}^{P}$$

**Definition** **3.***The execution time of*$t_{i}^{B}$. *According to the backup state of tasks, the execution time of the backup copy depends on whether its primary*$t_{i}^{P}$*succeeds. Assuming that the backup*$t_{i}^{B}$*is scheduled in the container*$c_{st}$*(**not in the edge node**of*$t_{i}^{P}$)*, then:*

(5)$$eq_{st}\left(t_{i}^{B}\right)=\left\{\begin{array}{c} 0,\\ f_{i}^{P}-s_{i}^{B},\\ \frac{l_{i}}{P_{st}\left(q\left(t_{i}^{B}\right)\right)}, \end{array}\right.\begin{array}{c} \eta_{ij}^{P}=1,state\left(t_{i}^{B}\right)=0\\ \eta_{ij}^{P}=1,state\left(t_{i}^{B}\right)=1\\ \eta_{ij}^{P}=0 \end{array}$$
where $\eta_{ij}^{*}$ denotes whether task copies succeed on $EN_{j}$. If so, $\eta_{ij}^{*}=1$; otherwise, $\eta_{ij}^{*}=0$. Equation (5) is explained as follows:
If $t_{i}^{P}$ succeeds and $state\left(t_{i}^{B}\right)=0$, then $t_{i}^{B}$ is not executed;If $t_{i}^{P}$ succeeds and $state\left(t_{i}^{B}\right)=1$, then the execution time of $t_{i}^{B}$ is $\left[s_{i}^{B},f_{i}^{P}\right]$;If $t_{i}^{P}$ fails, then the execution time of $t_{i}^{B}$ is $eq_{st}\left(t_{i}^{B}\right)=\frac{l_{i}}{P_{st}\left(q\left(t_{i}^{B}\right)\right)}$.


In order to meet the delay requirements of tasks and reduce the time redundancy between the active backup and its primary copy, the following conditions should also be met:(6)$$\forall t_{i}\in T,eq_{st}\left(t_{i}^{B}\right)\leq d_{i}-lst_{i}^{B}$$

When a task copy is successfully assigned to an edge container, the remaining processing capability of the container should be calculated first.

**Definition** **4.**
*Remaining processing capability. This is defined as the number of instructions processed by the remaining resources of the edge container per second, and the unit is MIPS. Therefore, the remaining processing capability of*
$c_{kl}$
*and*
$EN_{l}$
*are shown in Equations (7) and (8).*



(7)$$P_{kl}^{rem}=P_{kl}-\overset{N}{\underset{i=1}{\sum }}P_{kl}\left(q\left(t_{i}\right)\right)$$
(8)$$P_{l}^{rem}=P_{l}-\overset{m}{\underset{k=1}{\sum }}\overset{N}{\underset{i=1}{\sum }}P_{kl}\left(q\left(t_{i}\right)\right)$$


Maximizing the average QoS level of task copies in the edge container under fault tolerance is one of the scheduling objectives of this paper.

**Definition** **5.***The average QoS level. This is obtained as the ratio of all QoS levels of all successful tasks to the number of all successful tasks* [30]*, denoted by AQL. If*
$t_{i}^{*}$
*is successfully assigned to a container, then*
$z_{i}^{*}=1$*; otherwise,*
$z_{i}^{*}=0$.


(9)$$AQL=max\frac{\sum_{j=1}^{m}\sum_{i=1}^{n}z_{i}^{P}q\left(t_{i}^{P}\right)\eta_{ij}^{P}+\sum_{j=1}^{m}\sum_{i=1}^{n}z_{i}^{B}q\left(t_{i}^{B}\right)\eta_{ij}^{B}}{\sum_{j=1}^{m}\sum_{i=1}^{n}\left(z_{i}^{P}\eta_{ij}^{P}+z_{i}^{B}\eta_{ij}^{B}\right)}$$


The main goal of task scheduling in this paper is to make the edge container accommodate more task copies while adopting the PB model for fault tolerance. In the next section, we provide the solution according to the system model.

## 4. Solution: FASDQ Framework and Key Algorithms

We designed the FASDQ framework shown in Figure 4, which consists of two main parts: the primary/backup scheduling mechanism and the resource-adaptive adjustment mechanism. When a task request is received at the edge, the task is first replicated into two copies by the Copy Controller, and the task copies are assigned different QoS levels by the QoS Controller to change their execution times according to different reliability quality requirements to determine the optimal scheduling slot for the copies. The task copies in the queue are rescheduled by the Resource Controller after the resources of the backups are released so that the primary copies of the tasks can start as early as possible.

In addition to dynamic QoS-aware task copy scheduling, the FASDQ framework also includes a container resource-adaptive adjustment mechanism, which adaptively replenishes resources for tasks with insufficient processing capability during task execution to improve resource utilization in the edge node. When task copies cannot be scheduled at the edge, the tasks are handed over to the cloud for processing.

The following introduces the dynamic QoS-aware primary/backup scheduling mechanism and the container resource-adaptive adjustment mechanism at the edge.

### 4.1. Dynamic QoS-Aware Primary/Backup Scheduling Mechanism

In this paper, we consider a case in which the traditional PB model application is extended at the edge container side. FASDQ uses a backup overlapping technique to reduce redundancy in the task scheduling process and determines the schedulable interval of tasks in the edge container with dynamic QoS-awareness to find the best time interval for scheduling. This process is represented by the scheduling function.

According to the analysis in [17], when the primary copies of multiple tasks are not scheduled in the same nodes, their backups in the same virtual machine can overlap. In this paper, we also apply this rule to our scheduler mechanism. In order to ensure that no conflicts occur during the execution of task copies and achieve fault tolerance, we allow backup copies to overlap in time and space when scheduling two copies of a task. Task copies in the same container are subject to the following constraints: (1) the primary copy cannot overlap with other copies; (2) the redundant part of the active backup copy cannot overlap with other copies; (3) for the passive backup copy, if $EN\left(t_{i}^{P}\right)=EN(t_{j}^{P})$, then their backups cannot overlap.

The goal of the function scheduler $schedulable\left(t_{i}^{P},c_{jk}\right)$ is to designate schedulable time slots for $t_{i}^{P}$ in $c_{jk}$. The schedulable time slots need to be chosen so that $t_{i}^{P}$ starts as early as possible before its deadline in $\left[a_{i},d_{i}\right]$; the unschedulable time slots are those that are occupied by other primary/backup copies. Therefore, we traverse the schedulable time slot in $\left[a_{i},d_{i}\right]$ from left to right to find the earliest start time $eq_{jk}\left(t_{i}^{P}\right)$ of $t_{i}^{P}$ on $c_{jk}$ that satisfies Equation (4); otherwise, return +∞.

The goal of the function scheduler $schedulable\left(t_{i}^{B},c_{st}\right)$ is to designate schedulable time slots for $t_{i}^{B}$ in $c_{st}$. If $t_{i}^{P}$ is scheduled successfully, then the scheduling of $t_{i}^{B}$ will start. The schedulable time slots need to be chosen so that $t_{i}^{B}$ starts as late as possible within its deadline in $\left[a_{i},d_{i}\right]$, and this slot is not part of $EN\left(t_{i}^{P}\right)$; the unschedulable time slots of $t_{i}^{B}$ are the time slots occupied by its primary copies, the time slot occupied by the backup copy before $f_{i}^{P}$, and the redundant parts of active backup copies. As shown in Figure 5, $t_{1}^{B}$ and $t_{5}^{B}$ are passive backups, $t_{2}^{B}$ and $t_{3}^{B}$ are active backups, and $t_{4}^{p}$ is the primary copy. In other words, the unschedulable time slots of $t_{i}^{B}$ in $c_{21}$ are $\left[s_{1}^{B},f_{1}^{B}\right]$, $\left[s_{2}^{B},f_{i}^{P}\right]$, $\left[s_{3}^{B},f_{3}^{P}],\;\mathrm{and}\;[s_{4}^{B},f_{4}^{P}\right]$. Therefore, the schedulable time slot in $\left[a_{i},d_{i}\right]\;$is traversed from right to left to find the latest start time $eq_{lk}\left(t_{i}^{B}\right)$ of $t_{i}^{B}$ in $c_{\mathrm{lk}}$ that satisfies Equation (6); otherwise, return −∞.

Algorithm 1 describes the process of dynamic QoS-aware task scheduling for the primary copies in order to assign QoS levels to primary copies on demand to guarantee reliability. The main function of Algorithm 1 is arranging the processing capability of the edge nodes in descending order so that the primary copies are first assigned to the edge container with a higher processing capacity (lines 1–3). Then, it finds the earliest start time of $t_{i}^{P}$ by $schedulable\left(t_{i}^{P},c_{jk}\right)$ and a container satisfied time constraint in Equation (4); if $t_{i}^{P}$ is successfully scheduled, the label flag is set to be 1, otherwise, it is set to 0 (lines 5–14); if failed, the QoS level will be degraded by 1 until it becomes $q_{1}$, and the above operation will be repeated (line 15). If it is still not found, then the function resAdapt() is called (see Algorithm 3) (lines 17–23).
**Algorithm 1****.** Primary Scheduling in FASDQ**Input:** The set of tasks: T; QoS level: $Q=\left\{q_{1},q_{2,}\cdots ,q_{m}\right\}$; the set of ENs: $EN=\left\{EN_{1},EN_{2},\cdots EN_{n}\right\}$.**Output:**$\left\{c\left(t_{i}^{P}\right)\right\}$ // The container that can be scheduled for $t_{i}^{P}$ with the earliest start time.**Begin:**1**for each** new task $t_{i}\in T$, **do**2  flag=0;$\;est_{}^{P}=+\infty$;$c\left(t_{i}^{P}\right)=null$;$q\left(t_{i}^{P}\right)=q_{m}$;3  Sort ENs in descending order of processing capability;4  **while**
$q\left(t_{i}^{P}\right)\neq q_{1}$, **do**  // The QoS level of $t_{i}^{P}$ is initialized to the maximum value within time constraints.5    **for each**
$c_{jk}\in EN_{j}$, do6      Set the processing capability $P_{jk}\left(q\left(t_{i}^{P}\right)\right)$;7      $eq_{kl}\left(t_{i}^{P}\right)=\frac{l_{i}}{P_{kl}\left(q\left(t_{i}^{P}\right)\right)}$;8      $est_{i}^{P}=schedulable\left(t_{i}^{P},c_{jk}\right)$;9       **if**
$est_{i}^{P}\leq est_{}^{P}$**and**
$eq_{jk}\left(t_{i}^{P}\right)\leq d_{i}-est_{i}^{P}$, **then**10      //Select the container with the earliest completion time of the primary.11         $c\left(t_{i}^{P}\right)=c_{jk}$; $est^{P}=est_{i}^{P}$; flag=1;12         Calculate the remaining processing capability of $c_{jk}$ and $EN_{j}$;13     **end if**
14   **end for**15   **if**
flag==0, **then**
$q_{t}=q_{m-1}$; // The QoS level is degraded.16  **end while**17  **if**flag==0, **then**  //The existing container resources cannot meet the scheduling requirements.18    $c_{new}=resAdapt\left(t_{i}^{P}\right)$;19    **i****f**
$c_{new}\neq null$, **then**20     $c\left(t_{i}^{P}\right)=c_{new}$; // Create a new container.21     return $c\left(t_{i}^{P}\right)$;22
     **end if**
23  **else** Reject $t_{i}$;24**end for****END**

The time complexity of Algorithm 1 depends on the number of tasks K. Suppose initially that there are L active nodes with N containers of each node in the neighborhood of the user, and let the QoS level of the task copies be a random integer in (1, m). m and L are negligible in comparison with N and K. Thus, the time complexity of $schedulable\left(t_{i}^{P},c_{jk}\right)$ is O(NK). Then, Algorithm 1 comes to the function resAdapt(). We will analyze its time complexity in Section 4.2.

If $t_{i}^{P}$ is successfully scheduled, the scheduling of $t_{i}^{B}$ starts. Algorithm 2 describes the process of dynamic QoS-aware task scheduling for the backup. The main function of Algorithm 2 is sorting the edge nodes in descending order of remaining processing capability, except for the nodes in which its primary copy is located (lines 1–3). Then, the latest start time of $t_{i}^{B}$ that satisfies the time constraint in Equation (6) is found by $schedulable\left(t_{i}^{B},c_{lk}\right)$ in a container and the backup states are marked. If a container is successfully selected, the label flag is set to 1, otherwise, it is set to 0 (lines 5–18). If failed, the QoS level will be degraded by 1 until it becomes $q_{1}$, and the above operation will be repeated (line 19); if it is still not found, then the function resAdapt() is called (see Algorithm 3) (lines 22–28).
**Algorithm 2****.** Backup Scheduling in FASDQ**Input:** The set of tasks $T_{p}$ whose primary copy is successfully assigned to a container, $\left\{c\left(t_{i}^{P}\right)\right\}$; $Q=\left\{q_{1},q_{2,}\cdots ,q_{m}\right\}$; the neighboring edge nodes of task $t_{i}$: $EN\left(t_{i}\right)\subseteq EN$.**Output:**$\left\{c\left(t_{i}^{B}\right)\right\}$ //After the primary is successfully scheduled in a container, output the container for which the backup is scheduled.**Begin:**1**for each** backup task $t_{i}^{B}\in T_{p}$, **do**2  flag=0; $lst^{B}=0$; $c\left(t_{i}^{B}\right)=null$; $q\left(t_{i}^{B}\right)=q_{m}$;3  Sort $\left\{EN-EN\left(t_{i}^{P}\right)\right\}$ in descending order of the remaining processing capability;4  **while**
$q\left(t_{i}^{B}\right)\neq q_{1}$, **do** //The QoS level of $t_{i}^{B}$ is initialized to the maximum value within time constraints.5    **for each**
$EN_{l}\subseteq EN-EN\left(t_{i}^{P}\right)$, do6      **for each**
$c_{lk}\in EN_{l}$, **do**7        Set the processing capability $P_{lk}\left(q\left(t_{i}^{B}\right)\right)$;8        Calculate $eq_{lk}\left(t_{i}^{B}\right)$ using Equation (5);9        $lst_{i}^{B}=schedulable\left(t_{i}^{B},c_{lk}\right)$;10         **if**
$lst_{i}^{B}\geq lst^{B}$**and**
$eq_{st}\left(t_{i}^{B}\right)\leq d_{i}-lst_{i}^{B}$, **then**11        //Select the container for the backup copy with the latest start time.12          $c\left(t_{i}^{B}\right)=c_{lk}$; $lst^{B}=lst_{i}^{B}$; flag=1;13          Calculate the remaining processing capability of $c_{lk}$ and $EN_{l}$;14         **if**
$f_{i}^{P}\leq lst_{i}^{B}$, **then**
$state\left(t_{i}^{B}\right)=0$; //Mark the backup state.15         **else**
$state\left(t_{i}^{B}\right)=1$;16
        **end if**
17      **end for**18    **end for**19   **if**
flag==0, **then**
$q_{t}=q_{m-1}$; // The QoS level is degraded.20  **end while**21  **if**
flag==0, **then**
$c_{new}=resAdapt\left(t_{i}^{P}\right)$;  //The existing container resources cannot meet the scheduling requirements.22   **if**
$c_{new}\neq null$, **then**23    $c\left(t_{i}^{B}\right)=c_{new}$; // Create a new container.24    return $c\left(t_{i}^{B}\right)$;25
   **end if**
26  **else** reject $t_{i}$;27**end for****END**

Assuming that the number of the successfully scheduled primary is X, the time complexity of $schedulable\left(t_{i}^{B},c_{lk}\right)$, in the worst case, is O(NX). Then, Algorithm 2 comes to the function resAdapt(). We will analyze its time complexity in Section 4.2.

When the primary copy of the task succeeds and the corresponding backup copy is released, the vacant time slot can be used by other copies in the same container. If the primary can start earlier and the backup can start later within the time constraint, the redundant part of the active backup can be reduced, and the time performance of the primary copy that has not yet been executed is improved. Therefore, a new start time is chosen for the other copies in that edge container. The selection process is shown in Figure 6, where $\sigma_{i}^{B}$ is defined as the time required to release the backup.

Figure 6a shows the arrangement of each task copy in the container before $t_{1}^{p}$ succeeds. Figure 6b: After $t_{1}^{p}$ succeeds and $t_{1}^{B}$ is released, then $schedulable\left(t_{3}^{P},c_{21}\right)$ determines whether $t_{3}^{p}$ can be scheduled in $\left[f_{1}^{P}-\sigma_{1}^{B},s_{2}^{B}\right]$. If so, the new earliest start time $nest_{3}^{P}$ of $t_{3}^{P}$ is returned, and $state\left(t_{3}^{B}\right)$ is changed. In this example, $state\left(t_{3}^{B}\right)$ is changed from 1 to 0, reducing energy consumption. Figure 6c: $schedulable\left(t_{2}^{B},c_{21}\right)$ determines whether $t_{2}^{B}$ can be scheduled in $\left[s_{2}^{B},f_{3}^{P}\right]$, and then the new latest start time $nlst_{3}^{p}$ of $t_{2}^{B}$ is updated; this can also change $\mathrm{state}\left(t_{2}^{B}\right)$ from 1 to 0, which can effectively reduce the redundancy.

After the primary succeeds, the resources and time interval occupied by the backup are released. The rescheduling of task copies can result in the earlier completion of the primary copy within the time constraints, and the backup can start later, which effectively reduces the task delay.

### 4.2. Container Resource-Adaptive Adjustment Mechanism

In the above fault-tolerant scheduling process, the copy rescheduling mechanism eases the resource utilization of one container. In this subsection, we present the optimization of resource utilization across edge nodes, which means that edge containers can accept as many tasks as possible. However, if the edge container accepts multiple tasks over a period of time, which may lead to insufficient available resources for some tasks, then it will not be able to meet the latency constraint of task requests. To address the above situation, this paper proposes an elastic adjustment mechanism for container resources: i.e., when the current computational resources cannot meet the demand of task copies, the mechanism can: (1) add containers dynamically, (2) balance their workload with other containers, and (3) migrate containers to another edge node with sufficient processing capability. This process is shown by resAdapt().

Migration constraints. To ensure fault tolerance when migrating containers across edge nodes, the following constraints need to be satisfied during container migration:
(1)$c\left(t_{i}^{B}\right)$ cannot be migrated to the edge node of $t_{i}^{P}$, nor can $c\left(t_{i}^{P}\right)$ be migrated to the same edge node of $t_{i}^{B}$. Otherwise, two copies of the task are in the same edge node and it cannot be fault-tolerant.(2)If there is overlap between $t_{i}^{B}$ and $t_{j}^{B}$, $c\left(t_{i}^{P}\right)$ cannot be migrated to $EN\left(t_{j}^{P}\right)$.


The newly created container has a different processing capability and should meet conditions in terms of time:(10)$$a_{i}+t\left(c_{new}\right)+\theta_{new}+eq_{new}\left(t_{i}^{*}\right)\leq d_{i}$$
where $t\left(c_{new}\right)$ denotes the creation time of the new container, and $\theta_{new}$ is the ready time of the task in the new container.

resAdapt() is a container resource-adaptive adjustment algorithm, as shown in Algorithm 3.
The set ENti of running edge nodes near the user is sorted in ascending order of remaining processing capability pirem, and these nodes are traversed in turn and judged to be able to create new containers (processing capacity is denoted by pcnew) to accommodate the task copy (lines 1–7). If there is no suitable running node, some containers are migrated to a nearby running node to make space to create a new container (lines 8–11).If the creation of containers in the running node still cannot meet the scheduling demand of task copies, an unpowered node near the task copy (processing capacity is indicated by pENnew) starts up, and a new suitable container in this node is created (lines 13–18).
**Algorithm 3.** Container resource-adaptive adjustment algorithm resAdapt()**Begin**:1**for each**ENi⊆ENti, **do**//Create a new container in the running neighboring node.2  Sort active nodes ENi in ascending order by the remaining processing power;3  Select a new container cnew satisfying Equation (10);4  **if**
pirem>pcnew, **then** //Determine whether the remaining processing power of ENi can accommodate the new container.5    Create cnew in ENk;6    **return**
cnew;7  **else**8     Migrate c(pmin) in ENk to other nearby nodes with migration constraints; //Migrate the container with the least processing power in the target node to another neighboring node to accommodate the new container.9     **if**
ENk can accommodate cnew, **then**10       Create cnew on ENk;11       **return**
cnew;12    **end if**13  **end if**14**end for**15**for each**ENnew in nearby nodes, **do** //Enable a neighboring node in shutdown mode.16  **If**
pENnew>pcnew&&ai+θnew+tENnew+tcnew+eqnewti∗≤di, **then**17    Create cnew in ENnew;18    **return**cnew;19  **else return**
null;20**end for****END**

We now evaluate the time complexity of Algorithm 3. Firstly, it takes O(L) to create a new container to an active node, and it takes O(L) to migrate a container to another destination node. Hence, the time complexity of adding a new container by migration (see lines 8–12) is O(L^2^). For creating a new container to $EN_{new}$, the time complexity is O(1). Therefore, the complexity of resAdapt() is O(L + L^2^ + 1) = O(L^2^).

In conclusion, the time complexity of primary scheduling in FASDQ is O(NK + L^2^), and the time complexity of backup scheduling in FASDQ is O(NX + L^2^).

## 5. Experiments

### 5.1. Formatting of Mathematical Components

This section describes simulation experiments for latency-sensitive tasks in FASDQ on the EdgeCloudSim [31] platform, which is capable of modeling edge computing environments and provides a modular architecture that can support various key functions. Since the FASDQ proposed in this paper is mainly in an edge environment, the experimental framework was designed at the edge, as shown in Figure 7.

In this study, Kubernetes v1.8.3 was extended and deployed on the Centos 7.0 system. The experiments used fault injection to interrupt the execution of task copies. Four types of edge nodes and four types of containers were set up in EdgeCloudSim, and the processing capability of each edge node was set to 1000, 1500, 2000, and 3000 MIPS. The processing capability of the containers was 250, 500, 700, and 1000 MIPS, respectively, to evaluate the performance of FASDQ.

We want the edge containers to process task requests as quickly as possible. Therefore, we implemented the Task Scheduler and Resource Manager modules in the edge server for edge containers, which enable the containers to run a larger number of tasks in parallel.

In this paper, we compare FASDQ with the following solutions:(1)Fault-tolerant elastic scheduling algorithm (FESTAL) [17]: it can be added or removed certain virtual machines depending on the load state to optimize resource utilization while supporting fault tolerance in the cloud. In terms of using virtualization technology, the approach in this paper uses the more lightweight container virtualization technology. This experiment applied FESTAL to the edge side for comparative experiments to verify the effectiveness of FASDQ in terms of latency and resource utilization.(2)Fault-tolerance based QoS-aware scheduling algorithm (FTBQA)[19]: adjusts the QoS level of tasks and has an adjustment mechanism, which is invoked when the backup copy of a task in a compute node is deleted after the corresponding primary copy succeeds.(3)Non-Resource Adjustment (NONRAD): a variant of FASDQ that does not use the resource-adaptive adjustment mechanism.(4)Non-Backup Overlapping (NONBOL): a variant of FASDQ that lacks backup overlapping techniques to verify the impact of backup overlapping techniques on system reliability and QoS levels.

To study the performance under a more realistic scenario, we used an open dataset, YFCC100M [32], provided by Yahoo in 2014. YFCC100M contains 100 million media objects. There are approximately 0.8 million videos and 99.2 million images. Each object in YFCC100M contains several metadata, such as media source, owner name, camera, title, tags, and location [33]. We filtered the multimedia contents of 300 videos and 300 photos in the YFCC100M dataset. Multimedia content, namely, the number of tasks, is a random number in $\left[5000,400,000\right]$. At a given time, some of the multimedia devices are active, and the average number of active devices is 320. The distance of the edge base station was set to 1000 m, and a hexagonal structure was adopted to depict the range of the geographical location [34].

In order to more clearly observe the impact of different parameters on performance, experiments based on random synthetic workloads were conducted. The data traffic generated from each source point is related to total active users and is transmitted to the edge computing system as packets. The packet size is a random number in $\left[34,6550\right]$ and the unit is Byte. The average packet arrival rate is one packet per second per node. The bandwidth capacity between the users and the edge network is 1 Gbps [35].

Assume that the task arrivals at the edge server follow a Poisson distribution and that the average interval of task arrivals is 1/λ, uniformly distributed in $\left[\frac{1}{\lambda },\frac{1}{\lambda }+2\right]$ [17]. The task size is uniformly distributed in $\left[1\times 10^{5},2\times 10^{5}\right]MI$ with a deadline of $d_{i}=a_{i}+\max\{eq\left(t_{i}^{*}\right)\}+basedeadline$ [36], and basedeadline is the benchmark deadline, which is a random positive real number in $\left\{100,140,\cdots ,420\right\}$ in units of s, indicating the range of task delays. The edge server failure rate is uniformly distributed with a time unit of 10^−7^/h [14].

In each set of experiments, only one parameter was changed, and the others were left unchanged. Table 2 shows the parameters and their values.

### 5.2. Analysis of Experimental Results

#### 5.2.1. Evaluation of the Effectiveness of Fault-Tolerant Scheduling

In this experiment, the effectiveness of fault-tolerant scheduling of primary copies was evaluated by the task guarantee rate. The guarantee rate (GR) [17] is defined as the percentage of total tasks that are guaranteed to be completed successfully among all submitted tasks.
(1)The task arrival interval 1/λ value range is from 0 to 16 with a step of 2 s. The value is inversely proportional to the workload per unit time. Other parameters remain unchanged.(2)Base deadline is set from 100 to 420 with a step of 40 s. The higher the value of the base deadline, the more relaxed the task deadline constraint. The other parameters remain unchanged.


Figure 8a shows the variation in the guarantee rate with the task arrival rate of the five algorithms. The arrival rate decreases as the task volume increases, but all have higher guarantee rate (GR) values. Because the task arrival rate decreases, the workload decreases and the resources are more sufficient, so the GR value increases as the arrival rate decreases. Compared with FASDQ, NONRAD, and NONBOL, NONRAD has the lowest GR value because it does not use the resource-adaptive adjustment mechanism proposed in this paper, and when the task volume is large, the current resources cannot meet the demand of the task copy. NONRAD cannot expand resources on demand, resulting in tasks that do not start on time, thus missing deadlines. When a large number of task requests occur at the same time, FASDQ has a better effect on GR. This is because FASDQ can confirm the QoS level of task copies to reduce their execution time, and the resource adjustment mechanism can add new resources adaptively. FASDQ is thus able to handle more tasks under the same conditions. This is in line with our expectations.

Figure 8b shows the effect of the base deadline on the GR value. The data show that base deadline has a greater effect on the GR value because when the task is urgent, the system does not have time to adjust the available resources; therefore, task copies that arrive later are not able to meet the deadline. For several methods, when the latency is relatively lenient, the variation in GR values is usually stable because they implement fault tolerance and use overlapping technology. In particular, FASDQ can provide different QoS levels to tasks according to their different urgency levels to guarantee reliable service. Moreover, FASDQ extends the fault tolerance mechanism by considering the dynamic resource adjustment technique when resources are insufficient. Therefore, FASDQ performs better in terms of GR value compared with the other methods.

Figure 8c shows that the guarantee rate decreases as the number of tasks increases. Since failures are uniformly distributed, the number of tasks increases, leading to more failures, but all methods maintain a high guarantee rate, except for FTBQA. When the number of tasks increases, the mechanism can handle new requests by adjusting resources; while FTBQA has no resource-adaptive adjustment, it can guarantee a high GR because if there are no resources in the edge cloud to meet the task requirements, the task will be considered for scheduling in a remote cloud data center. With the increase in the number of tasks, FASDQ’s GR value remains above 94%, which proves the effectiveness of the proposed method once again.

In the proposed method, the latency is reduced by providing a QoS level to change the execution time of task copies, allowing the system to accommodate more task copies, so the effect of task latency can be better measured by the average reduced time. Figure 9 shows the results for the average reduction in time for the five methods. Both FASDQ and NONBOL, which adaptively adjust resources, significantly reduce the task latency with time constraints between 180 and 420 s, while the reduction in time is not significant between 100 and 180 s. This is due to the resource-adaptive adjustment mechanism, which is optimized to reduce resource consumption and improve resource utilization, resulting in the increased data-processing capability of edge servers and thus reduced task completion time. The difference in the latency results between FESTAL and FASDQ shows that providing QoS levels to tasks also improves their execution time. However, for tasks with short latency constraints, FASDQ cannot significantly reduce the task latency because of insufficient resources for adjusting the urgent-task system.

Because FASDQ dynamically assigns the QoS level to copies during scheduling, this experiment also evaluated the average QoS level (AQL) on the basis of 1/λ, base deadline, and the number of tasks. AQL is the average QoS level of all accepted tasks, and a higher AQL indicates a higher reliability performance of the framework. Let $q(t_{i}^{*})$ denote the QoS level of task copies $t_{i}^{*}$ with $0\leq q\left(t_{i}^{*}\right)\leq 1$ taking a value of $\left[0,0.1,0.2,\cdots ,0.9,1\right]$.

Figure 10a shows the variation in AQL as the task arrival rate decreases. As the task arrival rate decreases, the burden is reduced, the competition for resources decreases, and the task copies can be assigned higher QoS levels. Since FASDQ, NONRAD, NONBOL, and FTBQA can dynamically adjust the QoS level according to the demand of the copies when scheduling tasks, its average QoS level is higher than that of FESTAL. Compared with NONRAD and NONBOL, FASDQ has a higher average QoS level, which indicates that the combined application of the proposed method provides a higher QoS level for task copies, enabling the edge containers to handle more tasks in the same time interval.

Figure 10b shows that the more relaxed the task deadline, the better the scheduling of task copies, and the task copies can receive more reliable service guarantees. Therefore, the QoS levels are improved in all five methods. The AQL of FESTAL fluctuates more and is smaller than those of the other methods because it schedules without providing QoS levels to the copies on demand. As shown, the AQL of FASDQ can also exceed 0.6 for urgent task deadlines. This is because FASDQ can adaptively assign different QoS levels according to the urgency of the task. The more urgent the task, the higher the QoS level, and the more resources it consumes, resulting in fewer tasks being processed. However, FASDQ uses a resource adjustment mechanism, so its average QoS level can be higher than those in the other methods.

Figure 10c depicts the decreases in AQL as the number of tasks increases. As shown, when the number of tasks is between 5000 and 40,000, FASDQ can obtain a higher AQL than the other methods. The continuous increase in the number of tasks with a fixed number of available resources causes the average computational resources per task to become increasingly scarce, and the task copies obtain a lower QoS level. The results of comparing NONBOL and NONRAD show that the resource-adaptive adjustment mechanism proposed in this paper strongly improves the scheduling effect.

#### 5.2.2. Resource Consumption Evaluation

In this subsection, we verify the effect of the number of tasks on the total active time (HAT) of the edge nodes and the ratio of the service time of tasks to the active time of the edge node (RTH). The number of tasks in this experiment increased from 5000 to 40,000 with a step of 5000. HAT reflects the resource consumption of the system, RTH reflects the resource utilization of the system, and RTH is defined as follows:(11)$$RTH=\underset{t_{i}\in T,EN_{k}\in EN}{max}\left\{TET/HAT\right\}$$
where *TET* denotes the total execution time of the task.

Figure 11a shows that as the number of tasks increases, the HAT of the five algorithms maintains an increasing trend. The HAT is worse in both FTBQA and NONRAD, but for different reasons. FTBQA does not use container technology, and when there is a lack of computing resources, it needs to increase the edge server resources to meet task requirements, which results in significant resource consumption. The HAT is worse in NONRAD because there is no resource adjustment mechanism to replenish the processing capability for task copies in a timely manner, which also further illustrates the predominant role of resource-adaptive adjustment in fault-tolerant scheduling. When the number of tasks increases, FASDQ can migrate containers with a lower processing capability in the current candidate nodes to free up resources. This allows the candidate nodes to create new containers, thus avoiding the need to add or start new nodes and saving node resources.

Figure 11b depicts the variation in RTH performance as the number of tasks increases for the five methods. The RTH performance of NONRAD and FTBQA is poor when the number of tasks increases: when the tasks increase substantially, the system is burdened, and more active nodes must be added to the processing capability, resulting in greater resource consumption. The RTH of NONRAD outperforms that of FTBQA by 8.6%, indicating that the overlapping technique can effectively use the available resources to accommodate more tasks and avoid a significant increase in resources, effectively improving resource utilization. Compared with the other methods, the RTH of FASDQ is higher and relatively stable because FASDQ provides better QoS of reliability for task copies and reduces the execution time of tasks; therefore, FASDQ can always process more task copies in a given time, which effectively improves resource utilization.

In summary, the FASDQ mechanism extends the PB model to achieve fault tolerance by providing QoS levels for task copies, which effectively reduces task latency. From the evaluation of AQL, the FASDQ mechanism clearly provides higher QoS levels for latency-sensitive tasks and performs better in terms of reliability requirements and scheduling. As illustrated by the variation in HAT and RTH values, FASDQ can effectively improve resource utilization in the edge node and container levels by implementing container resource-adaptive adjustment in terms of consumption and saving resources.

## 6. Conclusions and Future Work

This paper introduces FASDQ, a fault-tolerant adaptive scheduling mechanism with dynamic QoS-awareness that guarantees reliable service for latency-sensitive tasks and improves the resource utilization of edge containers. FASDQ extends the PB model and proposes a dynamic QoS-aware task scheduling mechanism to achieve fault tolerance, which dynamically provides QoS levels to task copies according to the different latency requirements of tasks to change the execution time, so the edge containers can accommodate more task copies with a given processing capability while being fault-tolerant. In addition, FASDQ implements a resource-adaptive adjustment mechanism to address the uneven resource distribution and insufficient processing capability of tasks during execution, which effectively replenishes resources for tasks with insufficient processing capacity and reduces resource consumption at the node and container levels. The experimental results show that the method can significantly reduce the execution time of each task copy by about 12%, effectively reducing the task latency and improving system resource utilization.

In this paper, we do not consider task scheduling enough when the tasks are uploaded to the cloud after the failure of edge container scheduling, and we will further extend the fault-tolerant adaptive scheduling mechanism between cloud edges in the future so that the mechanism can tolerate multiple edge server failures. Furthermore, this paper addresses relatively static user requests, such as monitoring devices that are deployed in fixed locations. In realistic scenarios, such as 5G systems, there is a need to support mobility for mobile users. If the corresponding contents, which are located in edge containers, could be migrated to the place close to mobile users as they move, the QoS of reliability will be improved during the movement of the mobile users and the communication delay will be reduced between mobile users and the nodes. However, we did not delve into the task requests of the mobile users, which are common in realistic environments. This is our research focus for future studies.

## Figures and Tables

**Figure 1 sensors-21-02973-f001:**
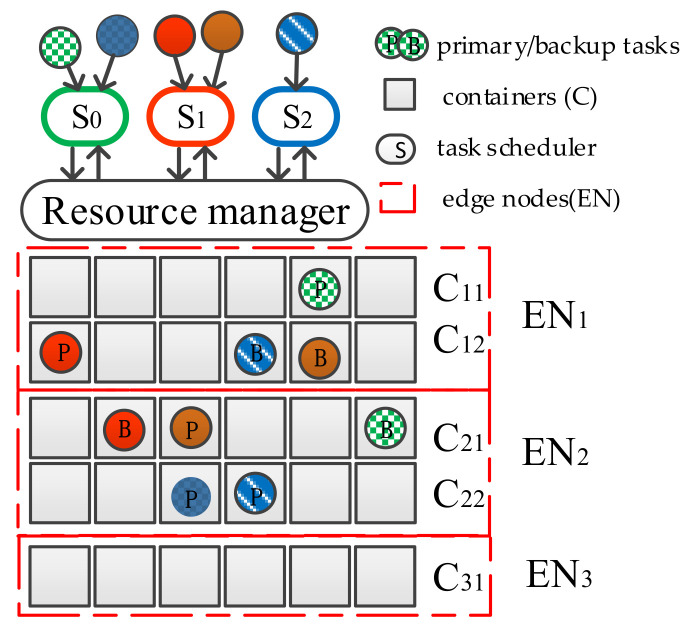
Problem analysis in the process of task copy scheduling.

**Figure 2 sensors-21-02973-f002:**
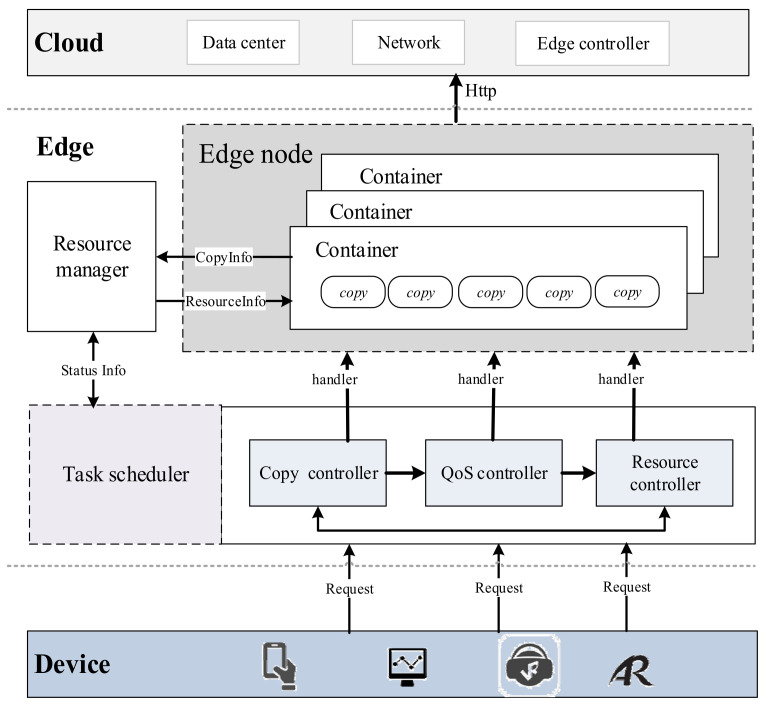
Task scheduling model, where QoS is Quality of Service.

**Figure 3 sensors-21-02973-f003:**
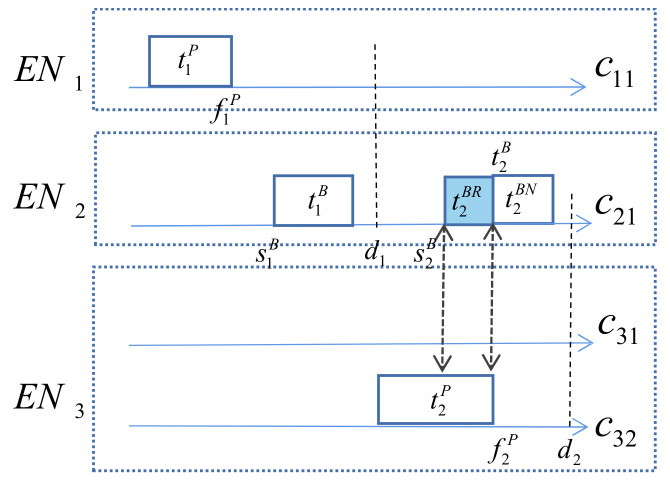
Time point of task copies.

**Figure 4 sensors-21-02973-f004:**
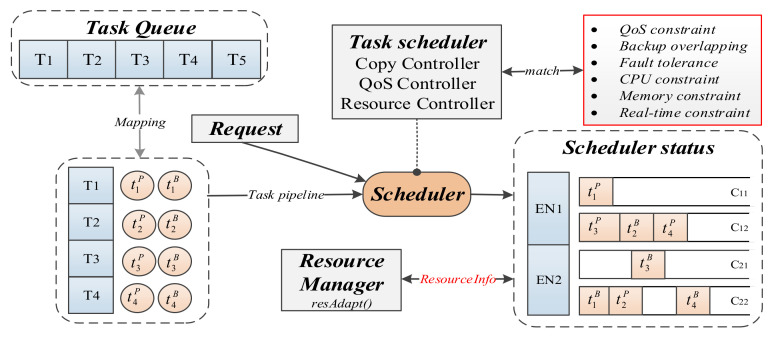
The framework of FASDQ (Fault-tolerant Adaptive Scheduling mechanism with Dynamic QoS-awareness).

**Figure 5 sensors-21-02973-f005:**
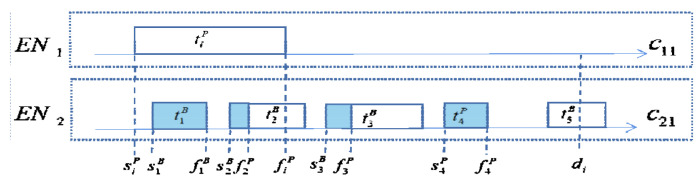
Unschedulable time slot for backup copies.

**Figure 6 sensors-21-02973-f006:**
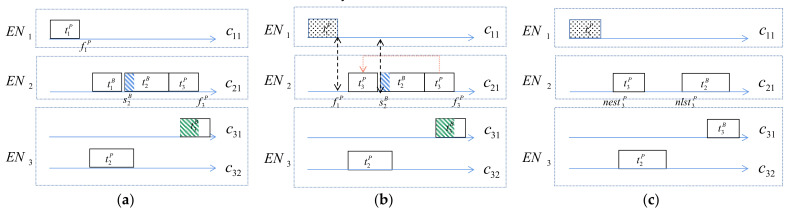
Task rescheduling process, is finished; (**b**) $t_{1}^{B}$ is released and $t_{3}^{p}$ is performed in advance; (**c**) $\mathrm{state}\left(t_{2}^{B}\right)$ is changed from active to passive.

**Figure 7 sensors-21-02973-f007:**
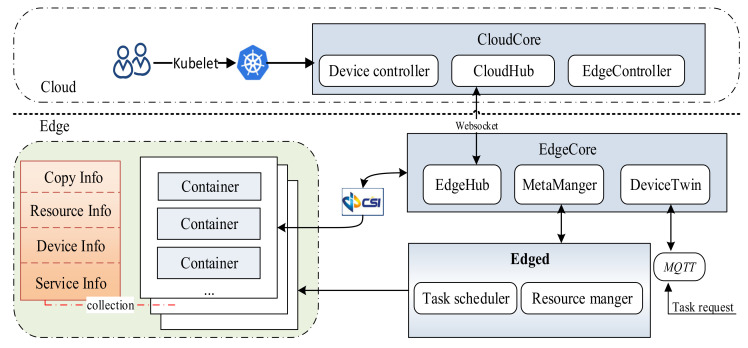
Task rescheduling process, where MQTT is message queuing telemetry transport.

**Figure 8 sensors-21-02973-f008:**
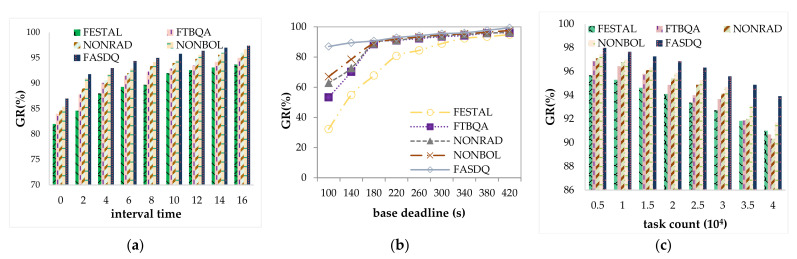
Comparison of guarantee rate (GR). (**a**) Impact of interval time on GR; (**b**) impact of base deadlines on GR; (**c**) impact of task count on GR.

**Figure 9 sensors-21-02973-f009:**
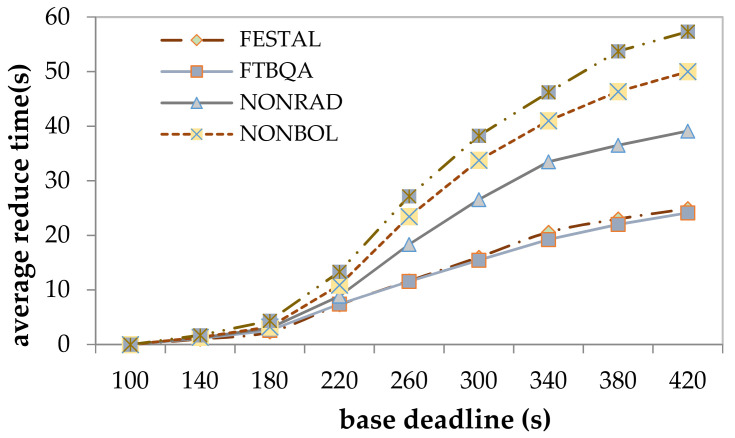
Average time reduction for a task.

**Figure 10 sensors-21-02973-f010:**
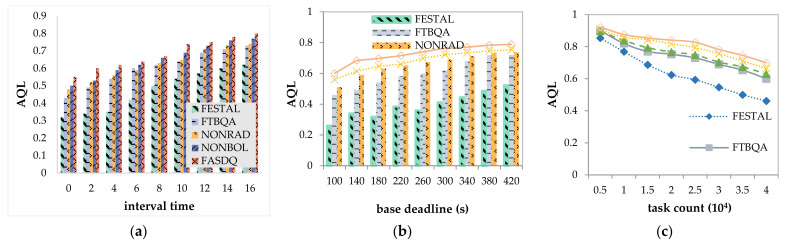
Comparison of average QoS levels (AQL). (**a**) Impact of interval time on AQL; (**b**) impact of base deadline on AQL; (**c**) impact of task count on AQL.

**Figure 11 sensors-21-02973-f011:**
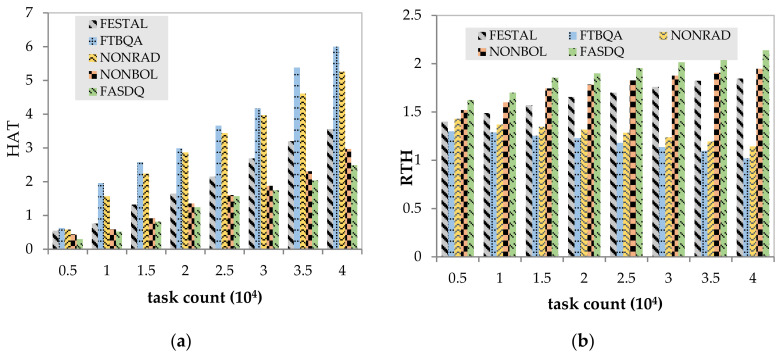
Comparison of resource utilization. (**a**) Impact of task count on HAT (the total active time of the edge nodes); (**b**) impact of task count on RTH (the ratio of the task service time to the active time of the edge node).

**Table 1 sensors-21-02973-t001:** Parameters and symbols.

Parameters and Symbols	Description
$EN_{i}$, $c_{ij}$	Edge node i and container j of edge node i.
$t_{i}^{*}$	Primary copy $t_{i}^{P}$ and backup copy $t_{i}^{B}$.
$a_{i}$, $\;d_{i}$	Arrival time and deadline of $\;t_{i}$ and their units are second.
$l_{i}$	Task size (unit: MI, millions of instructions).
$s_{i}^{P}$, $f_{i}^{P}$	Start time and finish time of $t_{i}^{P}$
$s_{i}^{B}$, $f_{i}^{B}$	Start time and finish time of $t_{i}^{B}$
$est_{i}^{P}$, $lst_{i}^{B}$	The earliest start time of $t_{i}^{P}$ and the latest start time of $t_{i}^{B}$
$state\left(t_{i}^{B}\right)$	State of $t_{i}^{B}$
$q\left(t_{i}^{*}\right)$	QoS level of $t_{i}^{*}$
$eq_{kl}\left(t_{i}^{*}\right)$	$t_{i}^{P}$
$eq_{kl}\left(t_{i}^{*}\right)$	The execution time of $t_{i}^{P}$ and tiB and the unit is second.
$P_{i}$, $P_{l}^{rem}$	The processing capability and the remaining processing capability of $EN_{i}$(unit: MIPS).
$P_{ij}$, $P_{kl}^{rem}$	The processing capability and the remaining processing capability of $c_{ij}$ (unit: MIPS).
$P_{kl}\left(q\left(t_{i}^{*}\right)\right)$	The processing capability provided to $t_{i}^{*}$ with $t_{i}^{*}$ of $c_{kl}$ (unit: MIPS).
$\eta_{ij}^{*}$	Indicates whether $t_{i}^{*}$ succeeds in $EN_{j}$: if it succeeds, $\eta_{ij}^{*}=1$; otherwise, $\eta_{ij}^{*}=0$
AQL	Average QoS level denoting the reliable demand of $t_{i}^{*}.$
$z_{i}^{*}$	Indicates whether $t_{i}^{*}$ is assigned to the container successfully.

**Table 2 sensors-21-02973-t002:** Task parameters.

Parameter	(Fixed)-(Min, Max, Step)
Task size (×10^5^ MI)	($\left[1,2\right]$)
Task count (×10^4^)	(1)-(0.5, 4, 0.5)
Interval time (1/λ)	(4)-(0, 16, 2)
Base deadline (×10^2^ s)	(3)-(1, 4.2, 0.4)

## Data Availability

The data presented in this study are openly available in YFCC100M at 10.1145/2812802, reference number [32].

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
