# Peer review of "FASDQ: Fault-Tolerant Adaptive Scheduling with Dynamic QoS-Awareness in Edge Containers for Delay-Sensitive Tasks"

_sensors, 2021, doi:10.3390/s21092973_

Round 1

Reviewer 1 Report

In this paper, the author proposed a fault-tolerant adaptive scheduling with dynamic QoS-awareness mechanism to reduce latency and improve reliability. This paper is good writing and clear. However, I have some question.

  1. the third one is not a contribution of the paper.
  2. Some abbreviations is not given the complete spellings, such as IIoT.
  3. There is no analysis of the proposed algorithm to demonstrate how the proposed algorithm to achieve the goal.
  4. If the proposed algorithm is used in practical scenario, what difficult it will encounter? Such as 5G, wireless sensor network.

Reviewer 2 Report

The manuscript deals with a scheduling algorithm for fault tolerance in edge cloud computing based on a simulation.

The core concept and main contributions are well defined and documented, but there are major issues that prevent publication.

  • Scientific representation is required. I recommend consulting an English editing service with a native speaker of English (extensive editing is needed).
  • The manuscript is poorly formatted and proofreading is required.
  • The evaluation is based on simulations. Thus, it is required to add details about realistic scenarios.
  • For evaluation, the authors should include their own insight and interpretations, not just describing the result itself.
  • More recent references are needed.
  • I suggest incorporating problem analysis and system model sections.

Reviewer 3 Report

The paper presents FASDQ, a fault-tolerant adaptive scheduling mechanism with dynamic QoS-awareness. Simulation results show that the proposed mechanism guarantees reliable service for latency-sensitive tasks and improves resource utilization of edge containers.

In some places the presentation is hard to follow due to typos and formatting mistakes and I encourage the authors to revise the paper from this point of view. Some considerations regarding this aspect are given below:

  • The notations in Figure 1 are given only partially. What does the P and B letter mean in the task representation?
  • PB is defined multiple times. All acronyms should be defined where they are fists used (e.g. FESTAL).
  • The beginning of Section 3 does not seem right. The first paragraph looks like an extended caption. However, this should be a description of the figure. Therefore the text "Figure 2" should not be bold and should not have a full stop after it. Maybe writing "In Figure 2, edge ..." 

  • Caption of Fig. 6 appears on other page than the figure itself.

Although the simulation results demonstrate the effectiveness of the proposed approach as compared to other ones, their presentation should be also improved. For example the legend in Fig. 8 a and c does not show all the represented data. This should be corrected. The same comment applies to the following figures, which seem to have incomplete legends. Where is FASDQ in Fig. 9, since the legend does not indicate it.

Round 2

Reviewer 2 Report

The authors revised the manuscript based on the previous review.

Thus, I recommend the manuscript for publication.